# SGR-Q: Saliency-Guided Rescaling for Large Language Model Quantization

## Abstract

The prohibitive computational and memory demands of Large Language Models (LLMs) necessitate quantization techniques. However, Post-Training Quantization (PTQ) methods suffer significant performance degradation at low bit-widths (e.g., 4-bit or lower), while Quantization-Aware Training (QAT) is resource-intensive and impractical for billion-scale models. Recent Quantization for Parameter-Efficient Fine-Tuning (Q-PEFT) approaches, integrating scale or adapter tuning with PTQ, offer a compromise but still face accuracy collapse and high costs under low bit-widths. We identify special phenomena that fine-tuning part of salient parameters (scales) can achieve a better performance than full parameter tuning for low bit-widths quantization, while the parameter tuning ratio is related to inflection point of the Cumulative Distribution Function (CDF) based on the Hessian matrix. Based on the observation, we introduce a simple yet effective method to identify salient weights that contribute more to representational fidelity, and accordingly propose a new quantization framework *Saliency-Guided Rescaling (SGR-Q)*. SGR-Q introduces sparsity-Hessian based identification of salient weight columns, and selectively fine-tunes their quantization scales while freezing other parameters. Our scheme by fine-tuning only part of the scale parameters can help retain more pretrained model's generalization ability than fine-tuning all the scale parameters. Extensive experiments validate the superior performance of SGR-Q comparing with PTQ and Q-PEFT methods across benchmarks. For instance, selectively tuning only the top-40% salient scales achieves 2.5% higher average accuracy on six commonsense reasoning datasets with 60% lower tuning costs compared to the state-of-the-art full-scale tuning method PEQA.

## 1 Introduction

The explosive growth of large language models (LLMs) such as LLaMA (Touvron et al., 2023a;b; Grattafiori et al., 2024; Dubey et al., 2024), DeepSeek (Liu et al., 2024a;b; Guo et al., 2025) and GPT-series (Brown et al., 2020; Achiam et al., 2023; Hurst et al., 2024) has unlocked unprecedented capabilities in diverse language tasks. However, these models are characterized by their extensive parameters, which pose significant challenges for memory footprint and bandwidth (Frantar et al., 2022; Dettmers et al., 2022).

Model quantization has emerged as a highly effective technology for compressing neural networks. Specifically, it reduces the model size of LLMs and substantially saves GPU memory consumption by drastically compressing model weights into compact integer representations (Dettmers et al., 2022). Current quantization techniques primarily fall into Quantization-Aware Training (QAT) and Post-Training Quantization (PTQ). While QAT incorporates quantization constraints during training for minimal error (e.g., BitNet b1.58 achieves near-lossless ternary quantization (Wang et al., 2023)), it requires full retraining from scratch. This makes QAT impractical for extremely large models (Chen et al., 2024). PTQ, conversely, quantizes pre-trained models directly without retraining (Frantar & Alistarh, 2023; Frantar et al., 2022; Lin et al., 2024; Xiao et al., 2023), offering significant speed and practical benefits (Zhu et al., 2024). However, standard PTQ methods like GPTQ (Frantar et al., 2022) and AWQ (Lin et al., 2024) focus on block-wise reconstruction and suffer severe performance degradation below 4 bits due to limited optimizable parameters and over-looked cross-block interactions (Chen et al., 2024).

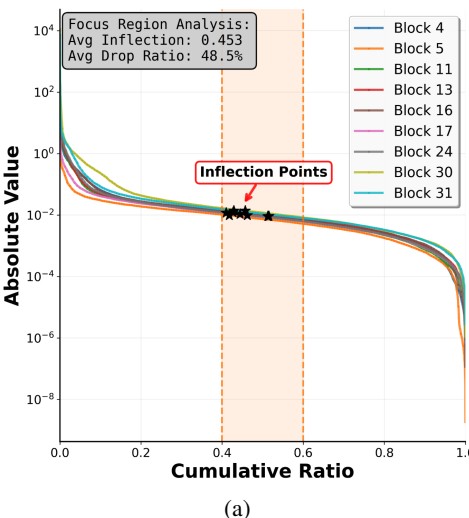 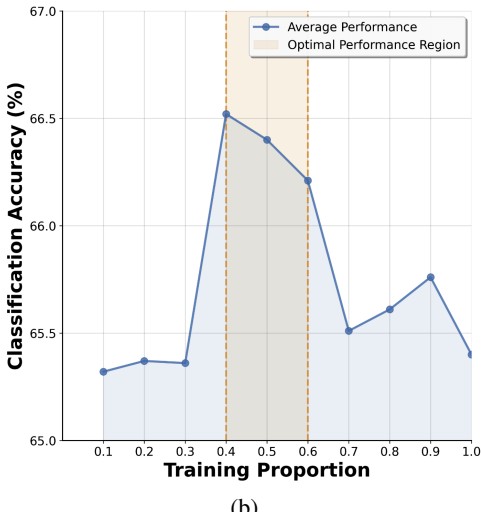

(a)                                                         (b)

Figure 1: Cumulative distribution analysis of the Hessian matrix. (a) The cumulative distribution curves of the Hessian matrix for each block of LLaMA2-7B, where the inflection points consistently fall within the 40%-60% ratio range, with a concentration towards the right of 40%. The x-axis and y-axis represent the cumulative ratio and the absolute value of the Hessian matrix, respectively. The black stars denote the inflection points of each block. (b) The relationship between the fine-tuning ratio of quantization scale parameters and model accuracy, showing that the highest average accuracy is achieved when the ratio is between 40% and 60%.

Recent research has explored Quantization for Parameter-Efficient Fine-Tuning (Q-PEFT) to address these limitations. Q-PEFT freezes quantized weights and fine-tunes minimal continuous parameters like quantization scales in PEQA (Kim et al., 2023) or adapters in QA-LoRA (Xu et al., 2023) and IR-QLoRA (Qin et al., 2024). This paradigm offers a promising compromise, providing additional adaptability beyond the rigid quantization constraints of standard PTQ. However, Q-PEFT methods still suffer from severe performance loss for ultra-low bit-widths. QA-LoRA needs to retrain a significant number of parameters with a high computational cost. In contrast, scale-tuning PEQA is more efficient, applying a uniform adjustment to all scales without considering their importance.

Notably, the existence of inherent weight saliency hierarchies in LLMs has been established, where a critical subset of weights disproportionately influences model performance (Lin et al., 2024; Shang et al., 2023; Huang et al., 2024). This insight aligns with AWQ's activation-guided preservation of salient weights (Lin et al., 2024). Similarly, GPTQ employs Hessian-based sensitivity analysis to identify and prioritize weights most vulnerable to quantization errors (Frantar et al., 2022). Analogically, mixed-precision methods like PB-LLM (Shang et al., 2023) explicitly allocate higher bit-widths to salient weights while compressing non-salient ones more aggressively. Nevertheless, utilizing saliency for directing the fine-tuning of quantization scales remains largely unexamined.

We first investigate the Cumulative Distribution Function (CDF) based on the Hessian matrix $H$ derived from the quantization loss function. As shown in Fig. 1(a), inflection points are observed near the 40% mark, revealing a consistent concentration of Hessian transition within this region. Based on the indication of weight importance by the Hessian matrix, we hypothesize that selectively fine-tuning the quantization parameters of the weights before the inflection point could enhance efficiency, since information loss becomes significant after inflection. We carry out ten groups of experiments with fine-tuning ratios ranging from 10% to 100%, in increments of 10% in Fig. 1(b). The results clearly demonstrate that fine-tuning within the 40%-60% range yields better performance, which confirms our hypothesis regarding the tuning ratio is related to the inflection point.

We accordingly design three parameter-efficient fine-tuning configurations according to Hessian-based importance and visualized their output feature differences from full-precision counterparts. As shown in Fig. 2, the results demonstrate that selectively tuning only the scales of non-salient weights fails to improve representational fidelity compared to full-scale fine-tuning and salient scale fine-tuning. This suggests that gradient noise may interfere with scale training beyond the inflec-

tion point. In addition, fine-tuning only part of the scale parameters retains more of the pretrained model's generalization ability than fine-tuning all the scale parameters. This phenomenon aligns with the *region-of-interest* (ROI) principle in compression theory (Wallace, 1991), where allocating resources to critical components maximizes fidelity.

Based on the observation above, we introduce ***Saliency-Guided Rescaling for efficient Large Language Model Quantization (SGR-Q)***. This new quantization fine-tuning framework leverages sparsity-Hessian guided saliency mask to fine-tune salient weights' scales exclusively. We compute sparse saliency representations through column-wise $L_1$-pooling of Hessian-based weight saliency, enabling targeted scale adaptation for salient quantized weight columns. Different from conventional full-scale fine-tuning, our method implements a three-stage precision repair protocol: initial PTQ for baseline low-bit representation, channel-wise Hessian screening to pinpoint top-k% critical weight columns, and importance-aware fine-tuning targeting only salient weight' scales while freezing non-critical ones. This strategy effectively mitigates the gradient noise dilemma by blocking potential noise propagation through frozen scales while permitting only fine-tuning for salient weights. In summary, the main contributions of this work are as follows:

- Through the analysis of Hessian-based importance and distribution deviation, we have verified that the weights, located before the inflection point of the CDF of the Hessian matrix, contribute more significantly to representational fidelity.

- We present a new fine-tuning framework based on Saliency-Guided Rescaling (SGR-Q), which can selectively fine-tune the scales of salient weights to enhance feature alignment in quantized LLMs, outperforming full-scale fine-tuning.

- Extensive experiments across low-bit quantization and binarization demonstrate that SGR-Q achieves better average accuracy than state-of-the-art methods, validating the effectiveness of our method.

## 2 RELATED WORK

Quantization is essential for scaling large language models by reducing memory and inference time.

**Post-Training Quantization (PTQ)** is the most practical approach for compressing models without retraining. Using block-wise second-order approximations, methods like GPTQ (Frantar et al., 2022) improve upon OBQ (Frantar & Alistarh, 2022). AWQ (Lin et al., 2024) and OWQ (Lee et al., 2024) enhance robustness with activation outlier detection. SmoothQuant (Xiao et al., 2023) and OmniQuant (Shao et al., 2023) address weight-activation quantization, using per-channel scaling and learnable clipping for low-bit flexibility. Despite progress, most PTQ methods ignore inter-block interactions and task-specific sensitivity (Frantar et al., 2022; Lin et al., 2024), leading to significant performance degradation at lower bit-widths. These limitations motivate the need for adaptive, structure-aware quantization strategies focusing on salient parameters.

**Quantization-Aware Training (QAT)** optimizes models directly under quantization constraints, enabling weights and activations to accommodate low-bit precision better. Although QAT is generally more accurate under ultra-low bit settings, its application to LLMs has been limited due to high training overhead. Recent work, including LLM-QAT (Liu et al., 2023) and BitDistiller (Du et al., 2024), incorporates knowledge distillation to mitigate instability during QAT. Methods like BitNet b1.58 (Wang et al., 2023) and OneBit (Xu et al., 2024) further extend QAT to binary and ternary regimes, achieving competitive accuracy on moderate-scale models. However, their scalability to larger LLMs remains uncertain due to the cost of full-model fine-tuning.

**Quantization for Parameter-Efficient Fine-Tuning (Q-PEFT)** combines quantization with Parameter-Efficient Fine-Tuning (PEFT) to adapt LLMs efficiently. Methods like QLoRA (Dettmers et al., 2023), IR-QLoRA (Qin et al., 2024), and LQ-LoRA (Guo et al., 2023) append LoRA modules to quantized models, but require merging during inference, which undermines compression. PEQA (Kim et al., 2023) avoids external modules by fine-tuning all quantization scales after PTQ, but suffers from instability due to indiscriminate updates. EfficientQAT (Chen et al., 2024) adopts a two-stage QAT: full-parameter training followed by scale-only adaptation. In contrast, our method SGR-Q introduces a saliency-aware, sparse fine-tuning strategy that selectively updates only the most impactful scales, identified via Hessian-guided screening. This enables robust adaptation with minimal overhead under low-bit settings.

## 3 PRELIMINARY

### 3.1 LOW-BIT QUANTIZATION FOR LLMS

For pre-trained weights of a fully-connected layer $\mathbf{W}_0 \in \mathbb{R}^{n \times m}$, we first perform round-to-nearest (RTN) quantization with $b$ bit width:

$$\widehat{\mathbf{W}}_0 = \mathbf{s}_0 \cdot \overline{\mathbf{W}}_0 = \mathbf{s}_0 \cdot \left( \text{clamp}\left( \left\lfloor \frac{\mathbf{W}_0}{\mathbf{s}_0} \right\rceil + \mathbf{z}_0, \, 0, \, 2^b - 1 \right) - \mathbf{z}_0 \right),$$

where the operation $\cdot$ indicates the element-wise product. The functions $\lfloor \cdot \rceil$ and $\text{clamp}(\cdot, \, l, \, r)$ indicate the rounding function, and the clamping function into the range $[l, \, r]$, respectively. Per-channel scales $\mathbf{s}_0 \in \mathbb{R}^{n \times 1}$ and zero-points $\mathbf{z}_0 \in \mathbb{R}^{n \times 1}$ are initialized to minimize $\|\mathbf{W}_0 - \widehat{\mathbf{W}}_0\|_F^2$. Here, we freeze $\overline{\mathbf{W}}_0$, which is the integer quantization indices of $\mathbf{W}_0$ for every fully connected layer in a pre-trained LLM, establishing a stable baseline for subsequent adaptation.

### 3.2 BINARIZATION FOR LLMS

Binarization is an extreme form of quantization, with the BiLLM (Huang et al., 2024) residual binarization framework being a typical PTQ method employed for LLMs. Specifically, BiLLM partitions weights into *salient* and *non-salient* components.

To accurately approximate salient weights using low-bit representations, BiLLM employs a two-stage residual binarization scheme:

$$\begin{cases} \alpha_o^*, \mathbf{B}_o^* = \arg\min_{\alpha_o, \mathbf{B}_o} \|\mathbf{W} - \alpha_o \mathbf{B}_o\|_2^2, \\ \alpha_r^*, \mathbf{B}_r^* = \arg\min_{\alpha_r, \mathbf{B}_r} \|\mathbf{W} - \alpha_o^* \mathbf{B}_o^* - \alpha_r \mathbf{B}_r\|_2^2, \end{cases} \quad (1)$$

where $\mathbf{W}$ represents the model's weights, $\mathbf{B}_o^*, \mathbf{B}_r^* \in \{-1, +1\}^{m \times n}$ are binary codes, and each scale factor $\alpha_o^*, \alpha_r^*$ is obtained via closed-form least-squares:

$$\alpha = \frac{\langle \mathbf{R}, \mathbf{B} \rangle}{\|\mathbf{B}\|_2^2}, \quad \mathbf{B} = \text{sign}(\mathbf{R}), \quad (2)$$

where $\mathbf{R}$ denotes the original weights $\mathbf{R}_o = \mathbf{W}$ or their residuals $\mathbf{R}_r = \mathbf{W} - \alpha_o^* \mathbf{B}_o^*$.

This two-level binary decomposition enables salient weights to be reconstructed with higher fidelity using only two binary matrices, thus avoiding full-precision storage. The final approximation of the salient weight block is given by:

$$\mathbf{W}_{\text{sal}} \approx \alpha_o^* \mathbf{B}_o^* + \alpha_r^* \mathbf{B}_r^*. \quad (3)$$

BiLLM applies a similar quantization strategy for non-salient weights after a sensitivity-based partitioning, but with more aggressive compression. Overall, this unified residual binarization framework allows BiLLM to achieve competitive performance under an average of just 1.08 bits per weight. However, the resulting binary residuals remain dense and fixed, limiting further compression and adaptive refinement opportunities.

## 4 EMPIRICAL STUDY

### 4.1 HESSIAN-IMPORTANCE ANALYSIS

Existing work has explored the relationship between the Hessian matrix of quantization loss and weight saliency, using the Hessian matrix as a key indicator of weight saliency (Frantar et al., 2022; Shang et al., 2023; Huang et al., 2024). However, the distribution of Hessian-guided saliency is still underexplored. As shown in Fig. 1(a), for LLaMA2-7B, we visualize the Cumulative Distribution Function (CDF) of the Hessian matrix for each block. We find that for nearly all layers and blocks, the Hessian distribution curve exhibits an inflection point in the 40%-60% range, with the concentration being highest just to the right of 40%. Since the Hessian matrix serves as an indicator of weight importance, we hypothesize that the critical weights lie before the inflection point, while the amount of information decreases significantly and the importance of the weights drops rapidly after the inflection point.

## 4.2 Quantization Distribution Deviation Analysis

Based on the observed Hessian inflection points above, we design three parameter-efficient fine-tuning configurations. Each configuration adjusts the quantization scales for a distinct set of weights selected by Hessian-based importance, corresponding to: (1) tuning only salient weight scales, (2) tuning only non-salient weight scales, and (3) tuning all scales. We then investigate the representational fidelity of different tuning strategies by visualizing the output feature distribution differences between full-precision and quantization computations. The proportion of salient weights is set to 40%. As shown in Fig. 2, the visual intensity of feature divergence shows a consistent hierarchy: Tuning only salient weights' scales yields minimal feature distortion, tuning all scales results moderate divergence, whereas tuning exclusively non-salient scales induces significant

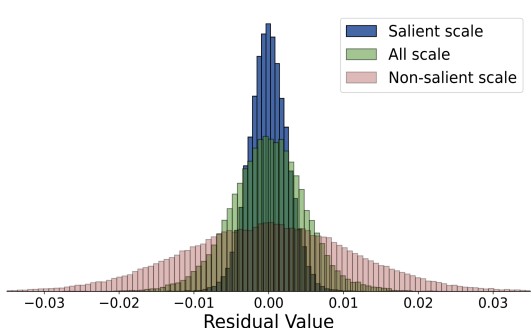

Figure 2: Quantization-induced feature distribution deviation relative to the full-precision model: Quantized models exhibit largest shift when tuning non-salient weight scales, and minimal shift when adjusting only salient weight scales.

distortion. In Appendix A.1, we also visualize the heatmap of this distribution. This visual progression demonstrates that selectively tuning only the scales of non-salient weights fails to improve representational fidelity compared to full-scale fine-tuning and salient scale fine-tuning.

## 5 Method

The analyses of Hessian-based importance and distribution deviation in the preceding section indicate that weights preceding the inflection point contribute more substantially to representational fidelity, justifying their prioritization during optimization. Conversely, adjustments to non-salient components risk introducing negative effects. **Building on this finding, we propose SGR-Q, which trains only the scales corresponding to the weights before the inflection point of CDF.** As shown in Fig. 3, SGR-Q implements a three-stage precision repair framework: (1) initial PTQ for baseline low-bit representations (as shown in Sec. 3 Preliminary); (2) sparsity-Hessian based saliency estimation to identify column-wise critical weights dominating quantization sensitivity; (3) importance-aware fine-tuning which exclusively updates quantization scales for top-k% salient weights while freezing all non-critical parameters. The following sections formalize our saliency metric design, selective optimization mechanism, and specific implementations for binarization.

### 5.1 Sparsity-Hessian Based Saliency Estimation

After initial PTQ based on Eq. (1), our approach leverages sparsity-Hessian based saliency measurement techniques to identify the most salient weights. Building on previous work in model compression (Frantar et al., 2022), we first compute the Hessian matrix $\mathbf{H}$ of the quantization loss function and a column-wise saliency metric $\mathcal{S}$:

$$\mathbf{H} = 2\mathbf{X}\mathbf{X}^T, \quad \mathcal{S}(w_{ij}) = \frac{|w_{ij}|^2}{[\mathbf{H}^{-1}]_{jj}^2}, \tag{4}$$

where $\mathbf{X}$ is the model's input activation, obtained from a small calibration set. This formulation extends standard Hessian-based pruning to the quantization domain, with $\mathcal{S}(w_{ij})$ combining weight magnitude ($|w_{ij}|^2$) and output sensitivity ($1/[\mathbf{H}^{-1}]_{jj}^2$).

Notably, quantized LLMs exhibit a structural dichotomy: A sparse subset of parameters disproportionately influences model performance, while the majority minimally influence task performance (Frantar et al., 2022; Lin et al., 2024). Therefore, we implement the influence estimation of weight through a column-wise saliency mask $\mathbf{M}$ derived from sparsity-Hessian based saliency scoring:

$$\mathbf{M}^* = [\mathcal{L}_1(\mathcal{S}_1), \mathcal{L}_1(\mathcal{S}_2), \ldots, \mathcal{L}_1(\mathcal{S}_G)], \tag{5}$$

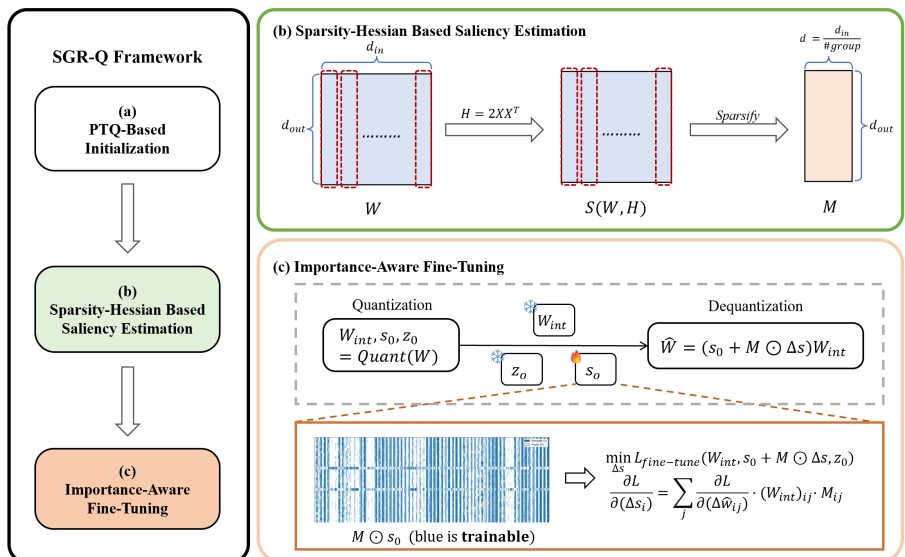

Figure 3: Overall Framework of SGR-Q. SGR-Q implemets a three-stage protocol: (a) PTQ-based initialization: establishes baseline low-bit representations; (b) Sparsity-Hessian based saliency estimation: computes weight significance using sparsity-Hessian approximation. Specifically, we compute sparse saliency representations through column-wise $L_1$-pooling of Hessian-based weight saliency; (c) Importance-aware fine-tuning: freezes scales of non-salient weight columns, exclusively fine-tunes scales of salient weight columns, therefore selectively update quantization scales.

$$\mathbf{M}_{ij} = \begin{cases} 1 & \text{if } \mathbf{M}_{ij}^* \in \text{top-}k\% \text{ of column } j \\ 0 & \text{otherwise}, \end{cases} \tag{6}$$

where $\mathcal{L}_1(\mathcal{S}_g)$ represents the channel-wise L1 norm for each group $g$ in the column-wise saliency matrix $\mathcal{S}$, and $k\%$ is a hyperparameter that determines the top percentage of weights selected based on saliency. Guided by mask $\mathbf{M}$, only the quantization scales of these selected weights are fine-tuned, while the scales of the remaining weights are frozen (*i.e.*, not updated via gradient descent). As validated in our ablation studies, this mask identifies critical weights where scale fine-tuning provides maximum benefit.

## 5.2 IMPORTANCE-AWARE FINE-TUNING

Capitalizing on the observation in our sparsity-Hessian guided saliency analysis (See Fig. 2), we eliminate performance deterioration from non-salient parameter adaptation in Q-PEFT via a sparse scale update strategy:

$$\widehat{\mathbf{W}} = (\mathbf{s}_0 + \mathbf{M} \odot \Delta\mathbf{s}) \cdot \overline{\mathbf{W}}_0, \tag{7}$$

where $\Delta\mathbf{s} \in \mathbb{R}^{n \times 1}$ is the scale adaptation term, and $\mathbf{M} \odot \Delta\mathbf{s}$ denotes Hadamard product with broadcasting, ensuring only salient scales receive updates. In other words, the scale training ratio is set to $k\%$ based on $\mathbf{M}$. This approach addresses the limitation identified in low-bit Q-PEFT by freezing non-salient scales and blocking noise propagation paths while preserving adaptation capacity in the most crucial parts.

The optimization minimizes loss with selective gradient flow:

$$\min_{\Delta s} \mathcal{L}_{\text{fine-tune}} \left( \widehat{\mathbf{W}} \right), \tag{8}$$

$$\frac{\partial \mathcal{L}}{\partial (\Delta s_i)} = \sum_j \frac{\partial \mathcal{L}}{\partial \widehat{w}_{ij}} \cdot \overline{w}_{ij} \cdot \mathbf{M}_{ij}. \tag{9}$$

The sparse gradient mechanism ($\mathbf{M}_{ij}$ term) ensures updates only flow through high-saliency weights, thereby preventing the feature distribution degradation when adapting non-salient scales. The full algorithmic details can be found in the Appendix A.2.

Table 1: Accuracy (%) of quantized LLaMA2 models under low-bit quantization in five-shot settings. All models fine-tuned on Alpaca dataset.

| Method | #Params | #Wbits | ARC-c | ARC-e | HellaSwag | PIQA | WinoGrande | Avg. |
|---|---|---|---|---|---|---|---|---|
| | | | **Five-Shot** | | | | | |
| LLaMA2 | 7B | 16 | 48.55 | 79.46 | 58.49 | 79.00 | 74.19 | 67.94 |
| | 13B | 16 | 54.18 | 81.99 | 61.38 | 79.92 | 76.09 | 70.71 |
| | 70B | 16 | 54.44 | 82.70 | 64.76 | 82.21 | 77.90 | 72.40 |
| +PEQA | 7B | 4 | 50.94 | 78.66 | 59.16 | 77.04 | 71.27 | 67.41 (-0.53) |
| | 13B | 4 | 53.75 | 81.44 | 62.16 | 80.14 | 73.48 | 70.19 (-0.52) |
| | 70B | 4 | 58.36 | 84.22 | 65.01 | 82.54 | 79.95 | 74.01 (+1.61) |
| +SGR-Q | 7B | 4 | 52.65 | 80.43 | 59.88 | 79.43 | 73.48 | 69.17 (**+1.23**) |
| | 13B | 4 | 55.89 | 83.54 | 62.80 | 81.18 | 76.95 | 72.07 (**+1.36**) |
| | 70B | 4 | 62.12 | 87.37 | 68.13 | 83.68 | 81.29 | 76.52 (**+4.12**) |

Table 2: Five-shot accuracy (%) on the MMLU dataset after the model is finetuned on the Alpaca dataset under low-bit quantization. The best and second best results are marked in bold and underlined receptively. For 13B models, our method achieves competitive performance with training time reduced to less than 1/10 of the best alternative.

| Method | #Params | Wbits | Humanities | Other | Social Sciences | STEM | Avg. |
|---|---|---|---|---|---|---|---|
| **LLaMA2-7B** | 7B | 16 | 43.27 | 52.20 | 51.74 | 36.04 | 45.81 |
| +PEQA | | 4 | 40.32 | 49.37 | 47.94 | 35.74 | 43.91 |
| +QA-LoRA | | 4 | 42.10 | 50.30 | 49.10 | 34.40 | 43.90 |
| +IR-QLoRA | | 4 | 43.40 | **53.60** | **51.90** | 36.80 | 46.20 |
| +SGR-Q | | 4 | **43.57** | 52.69 | 51.83 | **37.42** | **46.38** |
| **LLaMA2-13B** | 13B | 16 | 53.37 | 61.44 | 63.21 | 43.83 | 55.41 |
| +PEQA | | 4 | 50.42 | 57.62 | 60.30 | 41.66 | 52.51 |
| +QA-LoRA | | 4 | 48.00 | 57.40 | 59.70 | 43.00 | 51.70 |
| +IR-QLoRA | | 4 | 51.90 | **60.40** | **61.90** | **43.90** | **54.40** |
| +SGR-Q | | 4 | **53.05** | 59.58 | 61.65 | 42.28 | 54.00 |

**Binarization.** For binarization, we use BiLLM for initialization, along with the same saliency matrix computation. While BiLLM computes the scaling factors in Eq. (3) using closed-form least-squares, we observe that these static values may be sub-optimal when the quantized model is deployed on downstream tasks. To address this, we introduce lightweight fine-tuning of the two scaling parameters $\alpha_o$ and $\alpha_r$ only for salient weights, while keeping all binary codes $\mathbf{B}_o$, $\mathbf{B}_r$ fixed:

$$\min_{\Delta\alpha_o, \Delta\alpha_r} \mathcal{L}_{\text{fine-tune}}\left(W(\alpha_o + \Delta\alpha_o, \alpha_r + \Delta\alpha_r)\right), \tag{10}$$

$$\frac{\partial \mathcal{L}}{\partial \Delta\alpha_o} = \sum_{i,j} \frac{\partial \mathcal{L}}{\partial w_{ij}} \cdot \mathbf{B}_{o,ij} \cdot \mathbf{M}_{ij}, \quad \frac{\partial \mathcal{L}}{\partial \Delta\alpha_r} = \sum_{i,j} \frac{\partial \mathcal{L}}{\partial w_{ij}} \cdot \mathbf{B}_{r,ij} \cdot \mathbf{M}_{ij}, \tag{11}$$

where $W(\alpha_o, \alpha_r) = \alpha_o \mathbf{B_0} + \alpha_r \mathbf{B_r} + \mathbf{W}_{unsal}$ is the quantized weight, $\mathbf{W}_{unsal}$ denotes the non-salient weights and $\mathbf{M}$ denotes the saliency mask. All weights and non-salient scale remain frozen with original BiLLM quantization.

Our selective optimization exclusively fine-tune scales of salient weights and freeze remaining parameters, including non-salient scales and binary matrices, thereby minimizing the number of tunable parameters and reducing the risk of overfitting. Our experiments show that this strategy achieves better accuracy-efficiency trade-offs compared to full-scale fine-tuning or static quantization.

## 6 EXPERIMENT

We comprehensively evaluate SGR-Q across diverse settings, demonstrating its effectiveness in Quantization for Parameter-Efficient Fine-Tuning (Q-PEFT) and quantized model adaptation. SGR-Q outperforms existing Q-PEFT and PTQ baselines on multiple models and tasks with minimal scale updates. It further generalizes to binarization and these results highlight SGR-Q's practicality and versatility under resource constraints. According to Fig 1(a), we set the scale training ratio $k\%$ to

Table 3: Zero-shot performance of LLaMA family across different quantization methods under binarization on text-generation tasks and classification tasks. The gray row represents the full-precision model.

| Method | #W-Bits | Perplexity ↓ | | | Accuracy (%) ↑ | | | | | | |
|---|---|---|---|---|---|---|---|---|---|---|---|
| | | WikiText2 | PTB | C4 | BoolQ | HellaSwag | PIQA | WinoGrande | ARC-c | ARC-e | Avg. |
| FP | 16 | 5.68 | 41.15 | 7.34 | 75.11 | 56.94 | 78.67 | 70.01 | 41.89 | 75.25 | 66.31 |
| GPTQ | 2 | 3.2e3 | 2.9e4 | 7.7e4 | 45.47 | 25.85 | 52.01 | 48.30 | 23.55 | 25.42 | 36.77 |
| AWQ | 2 | 2.6e5 | 2.8e5 | 2.9e5 | 37.83 | 25.28 | 52.72 | 49.25 | 22.44 | 25.25 | 35.46 |
| QuIP | 2 | 21.22 | 231.06 | 20.02 | 52.94 | 36.93 | 62.51 | 55.41 | 23.04 | 40.45 | 45.21 |
| PB-LLM | $1.6_{(+1)}$ | 12.45 | 269.73 | 27.49 | 62.69 | 34.05 | 61.10 | 57.38 | 22.18 | 45.58 | 47.16 |
| BiLLM | 1.08 | 83.44 | 387.67 | 58.12 | 56.51 | 29.21 | 59.47 | 51.54 | 19.88 | 34.47 | 41.85 |
| +PEQA | 1.08 | 17.16 | 140.32 | 17.26 | 62.39 | 38.75 | 66.92 | 56.67 | 24.15 | 49.79 | 49.78 |
| +SGR-Q | 1.08 | **13.63** | **120.80** | **14.36** | **64.92** | **41.12** | **68.61** | **58.41** | **27.05** | **54.55** | **52.44** |

Table 4: Comparison of Training Strategies. We evaluate model performance on WikiText2 using perplexity.

| Method | Avg. Acc. ↑ | WikiText ↓ |
|---|---|---|
| Full-Precision | 64.83 | 5.47 |
| Train All Scales | 65.20 | 5.89 |
| Train Random 40% | 65.12 | 5.95 |
| Train No-Salient 40% | $64.70 \pm 0.07$ | 6.03 |
| Train Salient 40% | $\mathbf{66.52 \pm 0.02}$ | **5.73** |

Table 5: Comparison of fine-tuning methods in training time and trainable parameters on LLaMA2-7B and 13B.

| Method | #Bit | #Params (M) | | Time (h) | |
|---|---|---|---|---|---|
| | | 7B | 13B | 7B | 13B |
| QA-LoRA | 4 | 89M | 140M | 15.33 | 26.18 |
| IR-QLoRA | 4 | 89M | 140M | 15.40 | 26.20 |
| PEQA | 4 | 2.53M | 4.96M | 1.13 | 2.07 |
| SGR-Q | 4 | **1.02M** | **1.98M** | **∼1.12** | **∼2.05** |

40% in all experiments. Specific details regarding the impact of different training ratios are provided in the ablation study (Sec. 6.3 and Appendix A.3).

## 6.1 SGR-Q FOR LLM QUANTIZATION

**Implementation Details.** We conduct experiments on the LLaMA2 (Touvron et al., 2023b), OPT (Zhang et al., 2022) and Vicuna (Chiang et al., 2023) model families. Both our method and PEQA use group-wise quantization (group size = 128). All experiments were conducted on a single NVIDIA A800 GPU with 80 GB of memory. Additional training details and evaluation benchmarks are provided in Appendix A.4.

**Baseline.** As a baseline, we adopt PEQA (Kim et al., 2023), a representative post-training quantization (PTQ) plus fine-tuning method that further optimizes the quantization scale parameters during adaptation. In addition, recent LoRA-based quantization approaches, including QA-LoRA (Xu et al., 2023) and IR-QLoRA (Qin et al., 2024), are also compared. All methods are trained using the Alpaca dataset (Taori et al., 2023) described in Appendix A.4.

**Commonsense Reasoning Results.** Tables 1, A.4 and A.5 report our method's accuracy and perplexity results on the LLaMA2 model series. Compared to PEQA, which fine-tunes all quantization scales, our SGR-Q approach updates only a small subset yet achieves superior performance. On LLaMA2-7B/13B/70B, SGR-Q even outperforms the full-precision baseline by up to 1.69%, while consistently surpassing existing PTQ methods such as GPTQ (Frantar et al., 2022), OmniQuant (Shao et al., 2023), and AWQ (Lin et al., 2024). Regarding language modeling, our method achieves perplexity that closely matches the full-precision models. For instance, the gap on the C4 dataset with LLaMA2-70B is merely 0.06, indicating that SGR-Q retains strong generation capability despite aggressive quantization. While PEQA offers marginal improvements on larger models, we observe diminishing returns, likely due to gradient noise affecting specific scales during PTQ-based fine-tuning. Similar trends hold across the OPT and Vicuna models, where SGR-Q consistently delivers competitive or better accuracy. Results of 3-bit quantization and other models are provided in Appendix A.5 and A.6.

**Knowledge Reasoning Results.** To evaluate whether the fine-tuned model with SGR-Q can recover full-precision accuracy across a wide range of tasks after 4-bit quantization, we further conduct 5-

shot evaluations on the MMLU benchmark. We compare SGR-Q with PEQA and two LoRA-based fine-tuning methods, QA-LoRA and IR-QLoRA. As shown in Table 2, SGR-Q enables the LLM to regain its comprehension and in-context learning capabilities across diverse domains. Notably, on LLaMA2-7B, SGR-Q surpasses the full-precision baseline by 0.57%. While SGR-Q achieves comparable or marginally lower accuracy than LoRA-based approaches, it requires less than one-tenth the trainable parameters and training time of IR-QLoRA as shown in Table 5.

## 6.2 SGR-Q FOR LLM BINARIZATION

**Baseline.** Given that current 1-bit quantization typically incurs substantial accuracy degradation, we adopt BiLLM (Huang et al., 2024) as our baseline. We compare SGR-Q against representative post-training quantization (PTQ) methods, including GPTQ, AWQ, QuIP (Chee et al., 2023), and PB-LLM (Shang et al., 2023), as well as the fine-tuning-based approach PEQA, which is also built upon the BiLLM framework.

**Results.** As shown in Table 3, under the 1.08-bit setting, our SGR-Q method improves accuracy by 10.59% over the BiLLM baseline. Moreover, despite training significantly fewer parameters, SGR-Q outperforms PEQA by 2.66% while achieving notably better perplexity. Remarkably, SGR-Q even surpasses several 2-bit PTQ methods by a large margin, further demonstrating the effectiveness and practicality of our approach.

## 6.3 ABLATION STUDY

**Analysis of Training Scale Ratio.** We investigate the effect of the proportion of trainable scale parameters on model accuracy. Experimental results in Fig. 1(b) show that a 40% training ratio achieves optimal performance, corroborating our hypothesis on weight importance derived from the inflection point observed in Fig. 1(a). Complete experimental data are available in Appendix A.3.

**Effectiveness of SGR-Q Components.** To further verify the effectiveness of our SGR-Q design, we conducted an ablation study as shown in Table 4. We fine-tune different subsets of scale parameters, including: all scale parameters, 40% of the non-salient scales, 40% randomly selected scales, and our proposed 40% salient scales. As shown in Table 4, the best accuracy is achieved by fine-tuning only the salient scales, followed by all scales, random 40%, and the worst result comes from tuning only the 40% non-salient scales. In addition, we perform ten rounds of experiments by training on both the non-salient 40% and the salient 40% of scales, then compute the statistical mean and variance of the average accuracy. The results indicate that fine-tuning the scales of non-salient weights indeed leads to a significantly higher variance, specifically more than three times that of salient training, which further validates the presence of non-negligible gradient noise. This elevated gradient noise directly impairs the optimization stability of the non-salient components.

**Training Time and Number of Learnable Parameters.** As presented in Table 5, we report the training time and number of trainable parameters for four fine-tuning strategies across the LLaMA2-7B and 13B models. Our proposed SGR-Q method demonstrates significantly improved efficiency, requiring substantially less training time than LoRA-based methods, while also outperforming PEQA in terms of both time and parameter efficiency. Theoretically, our method is at least twice as fast as PEQA, as it leverages fewer trainable parameters and gradients. Despite tuning fewer parameters, SGR-Q achieves competitive accuracy, underscoring the effectiveness of selective adaptation in the low-bit quantization regime.

## 7 CONCLUSION

We proposed SGR-Q, a novel quantization adaptation framework that leverages sparsity-Hessian guided saliency estimation to selectively fine-tune the scales of the most salient weights in large language models (LLMs). Our experiments demonstrate that this selective approach achieves superior accuracy recovery and minimal feature distortion compared to full-scale tuning. SGR-Q significantly outperforms existing PTQ and Q-PEFT methods in low-bit and binary quantization, offering an efficient solution for quantizing large models. Future work will further optimize SGR-Q's hardware-accelerated training performance and apply SGR-Q to other architecture-based models.

## ETHICS STATEMENT

This research adheres to the ICLR Code of Ethics and follows ethical guidelines in all aspects of the study, including data usage and experimental procedures. All experiments were conducted using publicly available standard datasets (such as the MMLU dataset) and models (such as the LLaMA2 series), ensuring compliance with academic standards. This research does not involve human subjects or personal data.

We encourage the use of the technology presented in this paper for beneficial purposes, in accordance with ethical guidelines and legal standards.

## REPRODUCIBILITY STATEMENT

To ensure the reproducibility of our work, we provide all the necessary details required to replicate our results. The proposed SGR-Q framework, including Sparsity-Hessian Based Saliency Estimation and Importance-Aware Fine-Tuning, is described in detail in Section 1. Additionally, our experimental setup, including data, training hyperparameters, and other details, is fully described in the appendix and experimental sections. The data and models we use are publicly available and widely used in the community.

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

# A APPENDIX

## A.1 VISUALIZATION OF FEATURE DEVIATION

Fig. A.1 presents the output feature distribution differences from the full-precision (FP) model at the layer.6.self_attn.o_proj layer of the LLaMA2-7B model under different fine-tuning strategies. We design three parameter-efficient fine-tuning configurations, arranged from top to bottom: tuning only 40% salient weight scales, tuning only 40% non-salient weight scales, and tuning all scales. In the heatmap, lighter colors indicate smaller deviations from the original FP features, providing a visual comparison of the impact each strategy has on feature integrity. The fine-tuning effectiveness follows the order: Tuning only 40% salient weight scales yields the best performance, followed by tuning all scales, while tuning only 40% non-salient weight scales results in the worst performance.

## A.2 OVERALL ALGORITHM

---
**Algorithm A.1** Saliency-Guided Rescaling for Large Language Model Quantization (SGR-Q)

---
**Require:** Pre-trained model $\mathcal{M}$, bitwidth $b$, group size $g$, fine-tuning dataset $\mathcal{D}$
**Ensure:** Adapted quantized model $\widehat{\mathcal{M}}$
1: **for** each quantizable weight matrix $\mathbf{W}^\ell$ in model $\mathcal{M}$ **do**
2:    $\widehat{\mathbf{W}}_0^\ell, \mathbf{s}_0^\ell, \mathbf{z}_0^\ell \leftarrow \text{Quantize}(\mathbf{W}^\ell, b)$ {Eq. (1,2)}
3:    $\overline{\mathbf{W}}_0^\ell \leftarrow \widehat{\mathbf{W}}_0^\ell \oslash \mathbf{s}_0^\ell$
4:    $\mathbf{X}^\ell \leftarrow \text{ForwardActivations}(\mathcal{M}, \text{Subsample}(\mathcal{D}, 0.1\%), \ell)$
5:    $\mathbf{H}^\ell \leftarrow 2\mathbf{X}^\ell(\mathbf{X}^\ell)^T$ {Eq. (6)}
6:    $\mathcal{S}^\ell \leftarrow \text{ColumnSignificance}(\mathbf{W}^\ell, \mathbf{H}^\ell)$ {Eq. (7)}
7:    $\mathbf{M}^\ell \leftarrow \text{ColumnSignificanceMask}(\mathcal{S}^\ell, g, k_b)$ {Eq. (8,9)}
8:    $\Delta\mathbf{s}^\ell \leftarrow \mathbf{0}$ {Initialize sparse scale delta}
9:    **for** k in epochs **do**
10:      $(\mathbf{X}, \mathbf{Y}) \sim \mathcal{D}$ {Sample batch}
11:      $\widehat{\mathbf{W}}^\ell \leftarrow (\mathbf{s}_0^\ell + \mathbf{M}^\ell \odot \Delta\mathbf{s}^\ell) \cdot \overline{\mathbf{W}}_0^\ell$ {Sparse forward, Eq. (10)}
12:      $\hat{\mathbf{Y}} \leftarrow \text{ModelForward}(\mathbf{X}, \widehat{\mathbf{W}}^\ell, \text{layer} = \ell)$
13:      $\mathcal{L} \leftarrow \text{FinetuneLoss}(\hat{\mathbf{Y}}, \mathbf{Y})$
14:      $\nabla_{\Delta s^\ell}\mathcal{L} \leftarrow \text{SparseBackward}(\mathcal{L}, \mathbf{M}^\ell)$
15:      $\Delta s^\ell \leftarrow \Delta s^\ell - \eta_b \cdot \text{Adam}(\nabla_{\Delta s^\ell}\mathcal{L})$
16:    **end for**
17:    $\widehat{\mathbf{W}}^\ell \leftarrow (\mathbf{s}_0^\ell + \mathbf{M}^\ell \odot \Delta\mathbf{s}^\ell) \cdot \overline{\mathbf{W}}_0^\ell$
18: **end for**
19: **return** Adapted model $\widehat{\mathcal{M}} = \{\widehat{\mathbf{W}}^\ell\}$

---

We summarize the complete SGR-Q pipeline in Algorithm A.1. Given pre-trained weights $\mathbf{W}_0$, we first apply standard post-training quantization to obtain the initial low-bit representation $\widehat{\mathbf{W}}_0$, along with its associated scale $\mathbf{s}_0$ and zero-point $\mathbf{z}_0$. A small calibration set is then used to compute column-wise Hessian scores, which are combined with weight magnitudes to estimate saliency. Based on these scores, we construct a binary mask $\mathbf{M}$ from sparsity-Hessian based saliency scoring that identifies the top-$k\%$ most salient weight columns.

Only the quantization scales corresponding to these salient columns are updated during fine-tuning, while all other scales remain frozen. During training, sparse scale updates $\Delta\mathbf{s}$ are selectively applied via the mask $\mathbf{M}$, and gradients are backpropagated only through the active subset of scales. This targeted rescaling strategy enables efficient adaptation with low overhead while maintaining high accuracy by preserving fidelity in critical regions of the model.

## A.3 ANALYSIS OF TRAINING SCALE RATIO

In this section, we investigate the impact of the proportion of trainable scale factors on model performance. Using the LLaMA2-7B model, we conduct experiments by varying the proportion of trainable scale parameters from 0 to 1 in increments of 10%. As shown in Table A.1, the model

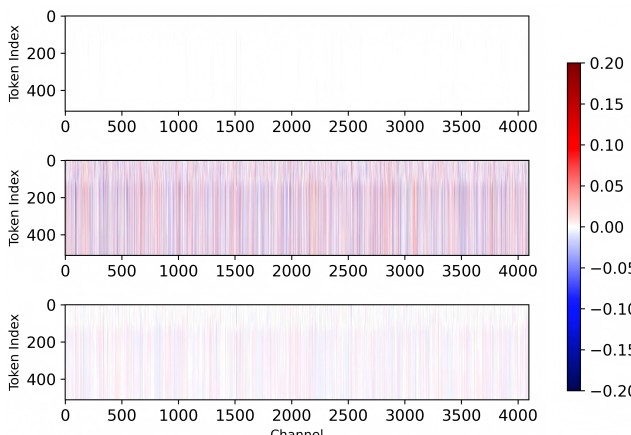

Figure A.1: Feature distribution differences from the full-precision model at the layer.6.self_attn.o_proj layer of the LLaMA2-7B model under three parameter-efficient fine-tuning configurations. From top to bottom: tuning only 40% salient weight scales, tuning only 40% non-salient weight scales, and tuning all scales.

Table A.1: Performance comparison of different proportion of high-bit (4bit) on LLaMA2-70B on five zero-shot classification tasks (accuracy, %).

| Training Proportion | #Params | ARC-c | ARC-e | HellaSwag | PIQA | WinoGrande | Avg. |
|---|---|---|---|---|---|---|---|
| FP | 7B | 43.52 | 76.35 | 57.13 | 78.07 | 69.22 | 64.86 |
| 100% | 2.53M | 47.27 | 76.94 | 58.43 | 76.77 | 68.61 | 65.40 |
| 90% | 2.28M | 46.16 | 76.85 | 57.13 | 78.89 | 69.77 | 65.76 |
| 80% | 2.02M | 47.10 | 76.85 | 57.91 | 76.99 | 69.22 | 65.61 |
| 70% | 1.77M | 45.14 | 76.85 | 57.37 | 78.51 | 69.69 | 65.51 |
| 60% | 1.52M | 47.10 | 77.65 | 58.63 | 77.80 | **69.85** | 66.21 |
| 50% | 1.28M | **47.35** | 78.08 | **58.66** | **78.84** | 69.06 | 66.40 |
| **40% (Ours)** | 1.01M | **47.35** | **78.62** | **58.66** | 78.45 | 69.53 | **66.52** |
| 30% | 0.76M | 44.71 | 77.36 | 57.39 | 78.67 | 68.69 | 65.36 |
| 20% | 0.51M | 45.22 | 77.57 | 57.21 | 78.24 | 68.61 | 65.37 |
| 10% | 0.25M | 44.97 | 77.78 | 56.98 | 78.45 | 68.43 | 65.32 |

achieves the best performance when approximately 40% of the scale parameters are trained. We observe a similar trend across other model variants, with 40% consistently yielding the optimal results. Moreover, most other training ratios yield performance between PEQA and SGR-Q, with a few cases performing worse than PEQA. Based on this observation, SGR-Q adopts 40% of the scale parameters for fine-tuning.

## A.4 EXPERIMENTAL SETTINGS

**Training Configuration.** Models were trained for one epoch using 4,096 samples from the Alpaca dataset (Taori et al., 2023) with a 2,048-token context window. Scale parameters used learning rates of 2e-5 for 3/4-bit quantization and 5e-5 for binarization settings. Regarding the fine-tuning ratio of scales, we adopt a 40% salient ratio under the 3/4-bit settings. In the binary case, due to the particularity of BiLLM, we only fine-tune the salient weights responsible for residual representation in BiLLM.

**Evaluation Benchmarks.** Our evaluation covers three dimensions:

- **Commonsense Reasoning:** Zero-shot and five-shot accuracy on HellaSwag (Zellers et al., 2019), PIQA (Bisk et al., 2020), WinoGrande (Sakaguchi et al., 2021), and ARC (Clark et al., 2018)

Table A.2: Zero-shot accuracy (%) of OPT and Vicuna with PEQA and SGR-Q (accuracy, %)

| Series | Model | Method | ARC-c | ARC-e | HellaSwag | PIQA | WinoGrande | Avg. |
|--------|-------|--------|-------|-------|-----------|------|------------|------|
| **OPT** | 6.7B | Full-Precision | 30.72 | 65.61 | 50.48 | 76.28 | 65.43 | 57.70 |
| | | +PEQA | 32.59 | 64.52 | 51.12 | 75.46 | 62.67 | 57.27 (-0.43) |
| | | +SGR-Q | 34.13 | 67.17 | 51.53 | 77.04 | 64.64 | 58.90 (+1.20) |
| | 13B | Full-Precision | 32.94 | 67.09 | 52.45 | 75.90 | 65.11 | 58.70 |
| | | +PEQA | 33.11 | 62.71 | 50.26 | 73.67 | 56.83 | 55.31 (-3.39) |
| | | +SGR-Q | 36.01 | 68.64 | 52.92 | 76.71 | 64.80 | 59.82 (+1.12) |
| | 30B | Full-Precision | 34.64 | 69.99 | 54.31 | 77.64 | 68.27 | 60.97 |
| | | +PEQA | 35.49 | 65.99 | 54.22 | 75.90 | 63.30 | 58.98 (-1.99) |
| | | +SGR-Q | 36.77 | 70.12 | 56.28 | 76.66 | 67.32 | 61.43 (+0.46) |
| **Vicuna** | 7B | Full-Precision | 43.26 | 75.63 | 56.46 | 77.26 | 69.46 | 64.41 |
| | | +PEQA | 44.37 | 76.26 | 54.88 | 78.89 | 69.85 | 64.85 (+0.44) |
| | | +SGR-Q | 47.27 | 77.48 | 57.26 | 77.69 | 69.06 | 65.75 (+1.34) |

Table A.3: Accuracy (%) of quantized LLaMA2 models under 3-bit quantization in zero-shot settings. All models fine-tuned on Alpaca dataset.

| Method | #Params | #Wbits | ARC-c | ARC-e | HellaSwag | PIQA | WinoGrande | Avg. |
|--------|---------|--------|-------|-------|-----------|------|------------|------|
| | | | **Zero-Shot** | | | | | |
| LLaMA2 | 7B | 16 | 43.60 | 76.35 | 57.13 | 78.07 | 68.98 | 64.83 |
| | 13B | 16 | 48.46 | 79.42 | 60.06 | 79.05 | 72.14 | 67.83 |
| | 70B | 16 | 54.44 | 82.70 | 64.77 | 82.15 | 77.98 | 72.41 |
| +PEQA | 7B | 3 | 42.83 | 75.76 | 56.16 | 77.75 | 68.11 | 64.12 (-0.70) |
| | 13B | 3 | 47.78 | 77.99 | 59.19 | 78.84 | 70.48 | 66.86 (-0.97) |
| | 70B | 3 | 54.63 | 82.08 | 64.34 | 82.59 | 77.35 | 72.20 (-0.21) |
| +SGR-Q | 7B | 3 | 43.77 | 76.73 | 56.79 | 77.31 | 68.51 | **64.62 (-0.21)** |
| | 13B | 3 | 49.15 | 79.17 | 59.97 | 78.94 | 69.77 | **67.40 (-0.43)** |
| | 70B | 3 | 54.52 | 82.83 | 63.89 | 82.05 | 78.22 | **72.30 (-0.11)** |

- **Language Modeling:** Perplexity on WikiText2 (Merity et al., 2016), PTB (Marcus et al., 1993), and C4 (Raffel et al., 2020), following standard protocols (Frantar et al., 2022; Lin et al., 2024; Zhu et al., 2025)

- **Knowledge Reasoning:** Massive Multitask Language Understanding (MMLU) (Hendrycks et al., 2020) benchmark, comprising 57 diverse subjects for comprehensive evaluation of factual knowledge and reasoning

## A.5 SGR-Q FOR OTHER SERIES MODELS

In addition to the LLaMA-2 series, we also conduct experiments on the OPT and Vicuna model families. As shown in Table A.2, SGR-Q achieves performance comparable to the full-precision (FP) models and consistently outperforms PEQA across all four model sizes.

## A.6 QUANTIZATION RESULTS FOR 3-BIT MODELS

The main text presents results for 4-bit and 1-bit quantization. This section provides the 3-bit quantization results in Table A.3, using the same training and evaluation setup as the 4-bit case. Our method remains nearly lossless at 3 bits and consistently outperforms PEQA, which fine-tunes all scales.

Notably, in the 2-bit quantization setting, selectively tuning only the salient scales is less effective. We attribute this to the excessive information loss caused by extremely low-bit quantization (sub-2-bit), where fine-tuning a small subset of parameters cannot fully compensate for the degradation. In contrast, our binary results remain strong, mainly due to BiLLM's residual-aware binarization mechanism. Using two binary matrices to represent salient weights introduces additional representational capacity, which outweighs the information loss (e.g., weight and gradient signals) in non-salient regions. As a result, selectively tuning scales in salient regions outperforms full-scale tuning in the

Table A.4: Perplexity comparison of quantized LLaMA2 models under 4-bit quantization on Wiki-Text2 and C4 across different model sizes and quantization methods. All models fine-tuned on Alpaca dataset.

| Dataset | Method | wbits | 7B | 13B | 70B |
|---|---|---|---|---|---|
| **Wiki** | FP | 16 | 5.47 | 4.88 | 3.32 |
| | +PEQA | 4 | 5.89 | 5.32 | 4.01 |
| | +SGR-Q | 4 | **5.73** | **5.11** | **3.64** |
| **C4** | FP | 16 | 6.97 | 6.47 | 5.71 |
| | +PEQA | 4 | 7.37 | 7.01 | 6.07 |
| | +SGR-Q | 4 | **7.25** | **6.70** | **5.77** |

Table A.5: Accuracy (%) of quantized LLaMA2 models under 4-bit quantization in zero-shot settings. All models fine-tuned on Alpaca dataset.

| Method | #Params | #Wbits | ARC-c | ARC-e | HellaSwag | PIQA | WinoGrande | Avg. |
|---|---|---|---|---|---|---|---|---|
| **Zero-Shot** | | | | | | | | |
| LLaMA2 | 7B | 16 | 43.60 | 76.35 | 57.13 | 78.07 | 68.98 | 64.83 |
| | 13B | 16 | 48.46 | 79.42 | 60.06 | 79.05 | 72.14 | 67.83 |
| | 70B | 16 | 54.44 | 82.70 | 64.77 | 82.15 | 77.98 | 72.41 |
| +PEQA | 7B | 4 | 47.27 | 76.94 | 58.43 | 76.77 | 68.61 | 65.20 (+0.37) |
| | 13B | 4 | 47.70 | 77.48 | 60.86 | 78.73 | 71.27 | 67.21 (-0.62) |
| | 70B | 4 | 53.67 | 82.32 | 63.87 | 82.26 | 77.11 | 71.85 (-0.56) |
| +SGR-Q | 7B | 4 | 47.35 | 78.62 | 58.66 | 78.45 | 69.53 | 66.52 (**+1.69**) |
| | 13B | 4 | 50.77 | 80.05 | 61.45 | 79.76 | 72.53 | 68.91 (**+1.08**) |
| | 70B | 4 | 56.83 | 83.75 | 65.88 | 82.81 | 77.19 | 73.29 (**+0.88**) |

binary case. In future work, we also plan to explore the application of SGR-Q under ultra-low bit quantization settings.

## A.7 USE OF LARGE LANGUAGE MODELS STATEMENT

In this study, large language models (LLMs) were solely used as writing aids to enhance language quality (e.g., grammar corrections, wording optimization, and minor formatting adjustments). All algorithms, analyses, results, and conclusions were entirely developed by the authors. LLMs did not contribute to the generation of technical content or decision-making. LLMs were not used for data creation or labeling, and no evaluation items were exposed. Draft snippets were provided as prompts, and all outputs were manually reviewed by the authors before inclusion.

