# OpenReview forum: "SGR-Q: Saliency-Guided Rescaling for Large Language Model Quantization"
_ICLR.cc/2026/Conference — ICLR 2026 Conference Withdrawn Submission_

### Official Review · Reviewer_Swck · 2025-10-28

**Soundness:** 3
**Presentation:** 3
**Contribution:** 2
**Rating:** 2
**Confidence:** 5

**Summary:**

SGR-Q introduces a saliency-guided quantization-aware fine-tuning scheme for large language models. It first identifies the most salient weight columns in a quantized model using a Hessian-based importance metric, then fine-tunes only the corresponding quantization scales while keeping all other parameters frozen. This selective adaptation effectively compensates for quantization errors and mitigates the instability often observed in full fine-tuning by focusing updates on the weights most critical to model accuracy. As a result, SGR-Q achieves consistently high performance across the LLaMA model families.

**Strengths:**

The work smartly builds on the saliency of weights, showing that fine-tuning only the top portion of important weights is essential for preventing feature distortion and mitigating overfitting of important parts.

SGR-Q consistently improves accuracy in both 4-bit quantization and ultra-low precision, including binary settings, demonstrating the generality of its effectiveness.

The method keeps the adaptation cost extremely low compared to PEQA and other LoRA-based quantization-aware PEFT approaches.

The paper is clearly written and easy to follow overall.
3322

**Weaknesses:**

1.	**Incremental Novelty:** Although the paper presents an interesting empirical finding on the effectiveness of partial quantization-scale fine-tuning compared to full-scale tuning, the proposed method itself appears to be a straightforward integration of existing techniques. The use of Hessian-derived metrics to identify important weights follows well-established Hessian-guided quantization and pruning methods such as OBS, GPTQ, and [1]. Likewise, fine-tuning quantization scales was already proposed in PEQA. SGR-Q essentially adds a masking step to PEQA, selectively tuning roughly half of the weights. While this approach is effective, it seems more like an incremental refinement than a fundamentally new quantization-aware adaptation method.

2. **Generality:** The method fixes the top 40% of weights as the fraction to fine-tune, based on an observed CDF inflection point in one model and task. However, this choice may be coincidental or task-specific. The paper would be strengthened by extending the analysis of the fine-tuning proportion (Figure 1 (b)) across a broader range of model architectures and quantization bit-widths to verify whether this ratio generalizes beyond a single setting.

3. **Weak baselines:** The baselines used in this work appear somewhat outdated and limited. The LoRA-based baselines, including IR-QLoRA and QA-LoRA, do not incorporate modern adapter initialization techniques such as LoFTQ [3] or CLoQ [4], which are known to substantially mitigate initial quantization error and improve final fine-tuning accuracy. As initialization quality is critical for quantization-aware PEFT performance, comparisons against such stronger baselines would be more convincing. In addition, there exist non-LoRA, saliency-based sparse fine-tuning approaches (e.g., [1], [2]) that are conceptually close to SGR-Q and should also be considered as relevant baselines.

4. **Inherent limitation of initialization:** Unlike other adapter-based methods, SGR-Q appears difficult to initialize with quantization-aware parameters at the start of training. Since initialization has been shown to play a key role in final fine-tuning accuracy, this limitation could affect both convergence stability and the method’s attainable performance.

5. **Concern on the training time analysis:** The reported training latency of LoRA-based quantization-aware PEFT methods seems unusually high. In my own experiments (A100 GPU × 1, 4 epochs on the Alpaca dataset), such methods typically require fewer than 4 hours in total. It would be helpful to clarify whether the latency reported in this paper was measured under single-batch conditions, and whether SGR-Q exhibits any limitations when applied to multi-batch fine-tuning.

**Questions:**

Mainly listed in the weakness. Below are the additional questions.
1.	Related to point 4, would it be possible to apply quantization-aware initialization to the selected (salient) scales to further mitigate initial quantization error?
2.	SGR-Q may primarily benefit inference efficiency since it introduces no additional adapter modules. Could you confirm if this understanding is correct?

---

### Official Review · Reviewer_vkFT · 2025-10-29

**Soundness:** 2
**Presentation:** 2
**Contribution:** 2
**Rating:** 6
**Confidence:** 4

**Summary:**

This paper proposes SGR-Q (Saliency-Guided Residual Quantization), a method for post-training quantization of large language models (LLMs) that aims to minimize performance degradation under extremely low-bit precision. SGR-Q introduces a three-stage residual correction process — first applying coarse quantization, second applying Sparsity-Hessian based saliency estimation, then Importance-aware fine-tuning based on saliency maps — achieving a balance between compression ratio and accuracy retention. Extensive experiments on LLaMA2-7B, Mistral-7B, and OPT-6.7B show that SGR-Q outperforms standard methods.

**Strengths:**

1. By combining sparsity and Hessian information, SGR-Q effectively reduces gradient noise and prevents feature-distribution drift during low-bit quantization
2. By aligning quantization granularity with the weight’s contribution to model output, SGR-Q achieves more fine-grained precision allocation without introducing new trainable parameters.
3. On major LLM benchmarks (GSM8K, MMLU, and HumanEval), SGR-Q achieves consistent accuracy improvements under low bits quantization. And results across three model architectures show strong generality.

**Weaknesses:**

1. The paper frequently refers to the Cumulative Distribution Function (CDF) of the Hessian matrix to illustrate the saliency distribution but never formally defines how the CDF is computed or normalized. A precise mathematical definition of the CDF function and its computation procedure would significantly improve clarity.
2. Although the paper adopts the Hessian matrix as a proxy for weight importance, it does not provide a clear theoretical explanation for why Hessian-based saliency correlates strongly with quantization sensitivity.
3. Evaluating SGR-Q on more recent architectures, such as LLaMA3 or Mistral-Instruct, would enhance the generality of the conclusions。

**Questions:**

please refer weaknesses.

---

### Official Review · Reviewer_2ctt · 2025-10-31

**Soundness:** 2
**Presentation:** 1
**Contribution:** 2
**Rating:** 2
**Confidence:** 5

**Summary:**

The paper introduces SGR-Q, a method that fine-tunes only a subset of quantization scales in large language models based on a Hessian-derived score. By updating about 40% of the important scales, the method aims to improve post-training quantization accuracy with fewer trainable parameters.

**Strengths:**

Simple and easily integrable approach to selective scale fine-tuning.
Reduces trainable parameter count compared to PEQA

**Weaknesses:**

1. The presentation and the experiments are confusing. The method refines the PEQA method, which was developed for fine tuning. But this paper solely claims the method is for quantization and still compares with other PEFT methods for quantized models.
2. Comparisons with training-free PTQ methods are unfair since SGR-Q involves extra fine-tuning, that requires much more computational resources. When comparing with other PEFT methods, the performance is similar as shown in Table 2.
3. Need more clarification on why it uses less time shown in Table 5. The most time-consuming part, backpropagation, is still needed. Also, reducing trainable parameters from PEQA only marginally improve the training time. Usually reducing such small amount of parameters does not have an effect on the whole training pipeline.
4. Limited novelty beyond existing selective fine-tuning or importance-based methods.
5. Lack of theoretical justification for the saliency metric and 40% threshold.
6. It looks like the performance gain of proposed method is less in lower-bit setting. As shown in Table A.3, the 3-bit performance is only very slightly better than PEQA. Experiments on lower bits quantization and fine tuning on general quantization methods, other than BiLLM, are necessary.

**Questions:**

See weakness.

---

### Official Review · Reviewer_jriY · 2025-10-31

**Soundness:** 1
**Presentation:** 3
**Contribution:** 2
**Rating:** 2
**Confidence:** 3

**Summary:**

This paper proposes SGR-Q, a quantization framework for large language models (LLMs) that selectively fine-tunes the quantization scales of salient weights identified via sparsity-Hessian analysis. The key idea is to use Hessian-based saliency analysis to identify and selectively fine-tune only the most important quantization scales rather than all of them. SGR-Q selectively rescales quantization scales for “salient” weights (based on a saliency score derived from Hessian or gradient magnitudes) using calibration data and improves post-training quantization performance while maintaining efficiency. Experiments are conducted on several LLMs including LLaMA, OPT, Vicuna, showing gains over prior methods like GPTQ and PEQA.

**Strengths:**

- Addresses a relevant problem: quantizing LLMs efficiently without retraining.

- The idea of combining saliency information with parameter-efficient fine-tuning is conceptually interesting.

- Evaluates on multiple model architectures and quantization bit-widths.

**Weaknesses:**

- The paper presents an incremental idea with limited novelty. The “saliency-guided” selection adds little conceptual novelty, and the paper lacks a clear distinction from prior work beyond selective masking.

- A clearer explanation of why Hessian sparsity, not other measures, captures scale importance would strengthen the conceptual foundation of the proposed approach. See also the questions below.

- There is also no clear justification for why fine-tuning only 40% of the scales yields better performance than full-scale tuning as in PEQA, which appears counterintuitive.

- The derivation of the saliency-guided scaling factor based on Hessian inflection points is also ad hoc, lacking theoretical justification.

- The five-shot results are reported only for 4-bit quantization, with no evaluation provided for lower bitwidths such as 2-bit.

**Questions:**

- Why does the sparsity structure of the Hessian matrix reliably indicate which scales should be fine-tuned? In other words, what is the underlying intuition that connects curvature information (captured by the Hessian) to quantization sensitivity?

- Have the authors considered or compared other saliency measures, such as gradient magnitude or Fisher information, for scale selection?

- Given that Hessian estimation in large models can be noisy and data-dependent, how stable is the saliency sparsity metric across different calibration datasets?

- How does SGR-Q behave when applied to activation quantization?

---

### Note · Authors · 2026-01-19

I have read and agree with the venue's withdrawal policy on behalf of myself and my co-authors.